

# Using Google Earth Engine to monitor co-seismic landslide recovery after the 2008 Wenchuan earthquake

Wentao Yang[1], Wenwen Qi[2], Jian Fang[3]

[1]Three-gorges reservoir area (Chongqing) Forest Ecosystem Research Station, School of Soil and Water Conservation, Beijing
Forestry University, Beijing, 100083, China.
[2]National Institute of Natural Hazards, Ministry of Emergency Management of China, Beijing, China
[3]School of Urban and Environmental Sciences, Central China Normal University, Wuhan, China.

*Correspondence to*: Wentao Yang (yang_wentao@bjfu.edu.cn)

**Abstract.** Earthquake-triggered landslides can pose serious threats to mountain communities by remobilizing and providing
loose materials for debris flows in post-seismic years. However, how long co-seismic landslides recover remains elusive due
to limited observations. Using vegetation dynamics, we studied surface recovery of co-seismic landslides induced by the 2008
Wenchuan earthquake from May 2008 to July 2019 for over 20,000 km$^2$. Landsat derived vegetation recovery on all co-seismic
landslides has been assessed based on the Google Earth Engine, a cloud-based computing platform. We found most co-seismic
landslides have been recovering after the earthquake but the spatial pattern is heterogeneous. The epicentre region with low
elevations along the bottom of the Min River valley has the best landslide recovery, whereas many landslides on the high
Longmen Mountain are poorly recovered ten years after the earthquake. These unrecovered hillslopes and gullies together with
widespread loose debris indicate that surface processes on high mountains may still active and may provide source materials
for debris flows, threatening communities at low elevations. To decipher possible mechanisms, we further analysed the
relations between landslide recovery and twelve influencing factors, including slope, pre-seismic vegetation condition,
landslide depth, landslide area, elevation, ground peak acceleration of the earthquake, aspect, slope curvatures, topographic
positions, mean annual precipitation, ground cohesion strength and vegetation types. We found elevation, topographic position
and pre-seismic vegetation condition are the most important factors that influence landslide recovery over all others. This work
also demonstrates the efficiency of the Google Earth Engine for continuously monitoring landslide dynamics over large areas.

## 1 Introduction

Large earthquakes trigger thousands of hundreds of co-seismic landslides (Xu et al., 2014), denude vast area of vegetation
(Cui et al., 2012), leave widespread unstable hillslopes, and have long-term impacts on landscape evolutions (Keefer, 1994;
Parker et al., 2011, Yang and Qi, 2017). After major earthquakes, co-seismic landslides are widely distributed (Xu et al., 2014)
and erosion from landslide surface can be massive (Sidle et al., 2011). Vegetation recovery on landslides plays a positive role
on post-seismic slope stability and can be used to indicate regional slope stability after major earthquakes (Chen et al., 2020;
Li et al., 2016; Yang et al., 2018a). How long will the scars of a major earthquake persist not only influence post-seismic



socioeconomic recovery (Huang and Fan, 2013) but also effects post-seismic regional erosion and orogenic mass balance (Marc et al., 2016; Parker et al., 2011).

Earthquake-triggered landslides commonly spread large spatial areas and investigations of landslides surface recovery at regional scale have been carried out by jointly using optical remote sensing images and field reconnaissance (Keefer, 2002;

Xu et al., 2014). Vegetation changes in remote sensing images are major features for landslide monitoring (Khan et al., 2013; Li et al., 2016; Lin et al., 2008; Mondini et al., 2011; Saba et al., 2010; Stumpf and Kerle, 2011). The 2008 Wenchuan earthquake triggered ~190,000 co-seismic landslides (Xu et al., 2014). After the earthquake, changes of landslide surfaces have been intensively studied at a few local areas. For example, in the epicentre area, the total area of landslides has been found decreasing linearly in the first five to eight years (Fan et al., 2018; Tang et al., 2016; Yang et al., 2017; Zhang et al.,

2014; Chen, 2020). The situation is similar in the lower Mianyuanhe watershed (Li et al., 2016) and the Hongxi watershed (Yang et al., 2015). These works on post-seismic landslide surface recovery have been carried out in very limited spatial area and observations in other parts of the earthquake-affected region is still missing, because high spatial resolution images used in these works are very expensive, and frequently influenced by bad weathers. Lack of observations on landslide changes over the entire region hinders a holistic understanding of its evolving patterns and driving factors.

To overcome incomplete observations of post-seismic landslides after the Wenchuan earthquake, MODIS data with large footprints and short revisit time has been used (Yang and Qi, 2017; Yunus et al., 2020). The 250 m resolution MODIS data is sensitive to changes of landslide surfaces after the Wenchuan earthquake (Liu et al., 2015; Zhang et al., 2018). Long-term monitoring of post-seismic landslide surface using the MODIS time series revealed a spatially heterogeneous pattern (Yang and Qi, 2017). The recovery of MODIS derived landslide surface is found sensitive to precipitation and topography (Yang et

al., 2018b). Despite MODIS observation can monitor the entire earthquake-affected region, its spatial resolution is much too coarse and most signals of the 250 m MODIS are a mixture of landslides and other ground features.

Landsat imagery is a valuable data for monitoring long-term earth surface processes at a spatial resolution of 30 m for multi-spectral bands. The finer spatial resolution of Landsat over MODIS makes it possible to map a single medium sized landslide (e.g. 30m×30m) and it has been frequently used to map regional landslides (Behling et al., 2016; Chen et al., 2019; Coe et al.,

2018; Marc et al., 2015). Landsat 5 TM, Landsat 7 ETM and Landsat 8 OLI all have 16-day revisit interval and joint use of them lead to more frequent observations and easy comparisons for monitoring landslide surface dynamics. The short revisit time in time series of all Landsat images have the potential to overcome partial coverage problems encountered by very high spatial resolution images. Despite Landsat images also face the problem posed by clouds, some algorithms have been developed to minimize the influence of clouds (Zhu and Helmer, 2018; Zhu and Woodcock, 2012). To obtain cloud-free

observations, many Landsat images of different dates is needed and handling these large amounts of Landsat images could pose another challenge.

In this work, we explore a cloud-based platform, the Google Earth Engine (GEE), to map surface recovery of co-seismic landslides triggered by the $M_W$ 7.9 Wenchuan earthquake in an area of 23,000 km$^2$. This cloud-based platform has the advantage of easy-to-use and can efficiently process large volumes of data by researchers that are not familiar with remote



sensing image processing skills (Gorelick et al., 2017). The objectives of this work are: 1) to develop a method to map landslide recovery pattern by using large volumes of Landsat images, and 2) to explore possible factors that influence landslide recovery.

## 2 Methodology

### 2.1 Study area

The $M_W$ 7.9 Wenchuan earthquake occurred on May 12, 2008, in the Longmen Mountain, bordering the Tibetan Plateau
(average elevation >4000 m) and the Sichuan Basin (<800 m) (Fig. 1). This earthquake triggered nearly 200,000 landslides over 110,000 km² and the total landslide area is 1,160 km² (Xu et al., 2014). The size of these landslides varies over several orders of magnitudes, with the smallest a few square meters and the largest landslide up to ~8 km² (Hu et al., 2019). Distributed along the >200 km Yingxiu-Beichuan Fault ranging from the epicentre Yingxiu town north-eastward to Qingchuang county, these landslides formed a densely distributed landslide zone (>10% surface area disturbed by landslides) and the width of the
southern section (~25-30 km) is much larger than the northern part (~3-5 km) (Ouimet, 2010).

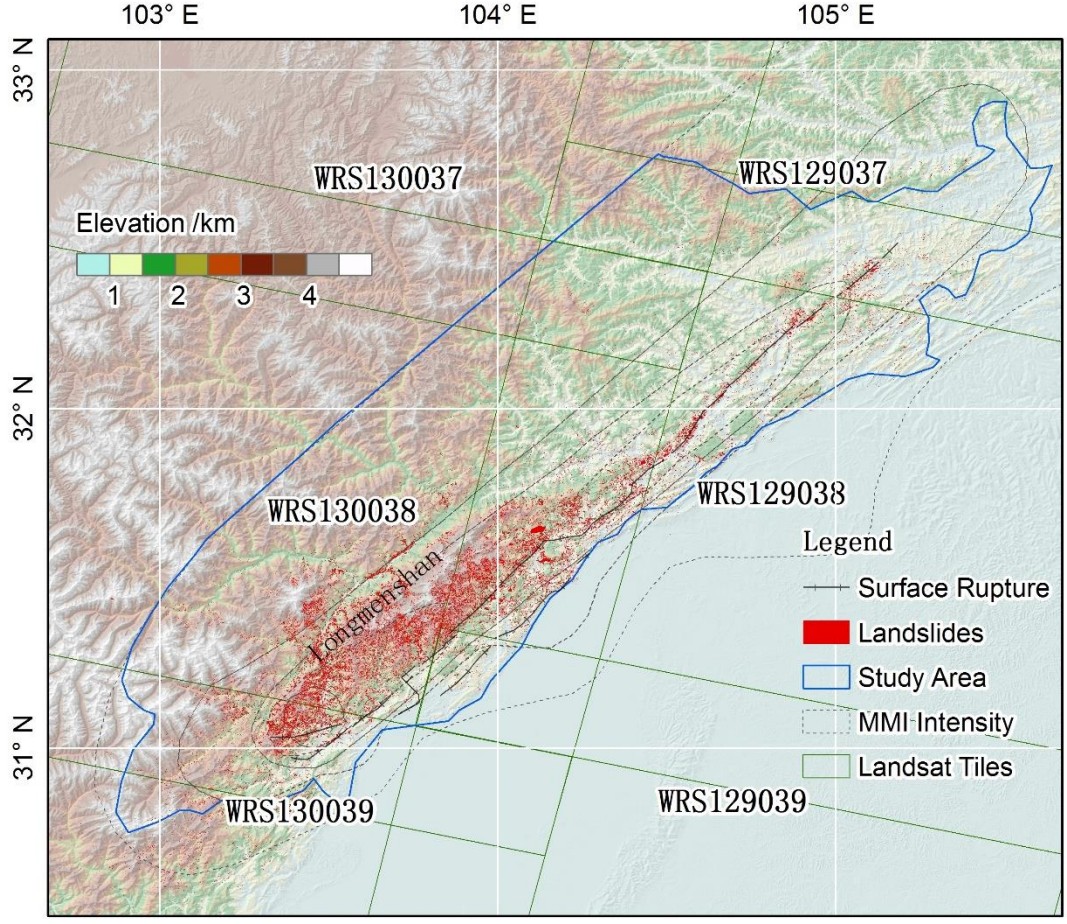



**Figure 1: Study area and Landsat tiles used in this work. Landslides triggered by the 2008 earthquake are interpreted by Xu et al. (2014). The shown DEM is SRTM DEM downloaded from the USGS website. The surface rupture is digitized from the map in Xu et al. (2009). The MMI intensity lines are produced by the China Earthquake Administration and were digitized by the authors.**

**2.2 Method**

This study area covers six Landsat Tiles: WRS129037, WRS129038, 129039, WRS130037, WRS130038, WRS130039 (Fig. 1). WRS129038 and WRS130038 are two major tiles that cover most part of this study area. In this work, we used Tier 1 calibrated top of atmosphere (TOA) reflectance product of the USGS Collection 1. At this level, the data is created using the best processing level (Chander et al., 2009). All Landsat 5, 7 images from 2001 to 12 May 2008 and all Landsat 5, 7, 8 images

from May 2008 to July 2019 are used to study pre- and post-seismic vegetation dynamics on earthquake-triggered landslides, respectively. There are 1,167 pre-seismic images and 1,857 post-seismic Landsat images used in this work. All image processing was performed in Google Earth Engine (Gorelick et al., 2017).

Previous works show that landslides can dramatically decrease vegetation index and the recovery of vegetation index to pre-seismic level can be used to indicate the time of landslide surface recovery (Yang et al., 2018a; Zhang et al., 2018). In this

work, we used enhanced vegetation index (EVI) to measure vegetation recover on co-seismic landslides, because EVI is more consistent than other vegetation indexes (such as NDVI) among three types of Landsat images (Bell et al., 2018; Zhu et al., 2016).

$$EVI = \frac{2.5*(\rho_{nir}-\rho_{red})}{\rho_{nir}+6*\rho_{red}-7.5*\rho_{blue}+1} \quad (1)$$

where $\rho_{nir}$, $\rho_{red}$, $\rho_{blue}$ are near infrared (760-900 nm), red (630-690 nm) and blue (450-510 nm) bands of Landsat TM/ETM+/OLI

images, respectively.

EVI time series after the earthquake from May 2008 to July 2019 were composed in a chronological order. Clear observations (not contaminated by clouds or snow) for each pixel were used to interpolate contaminated observations by an ordinary least square (OLS) regression method (Zhu and Woodcock, 2014; Zhu et al., 2019). Landslide surface recovery were assed based on changing trends of post-seismic EVI.

**2.2.1 Clear observation selection**

To select high-quality observations, we first used the Fmask in the TOA product to mask all possible low-quality pixels, including clouds and its relating shadows and scan-line corrector (SLC)-off gaps (Zhu et al., 2015; Zhu and Woodcock, 2012). The quality of pixels was marked by the Fmask and it can be used to remove most clouds, cloud shadows and circus clouds except some thin clouds and haze, which can be mistakenly regarded as EVI drop by landslides. To remove these low-quality

pixels left by the Fmask, we used a simple Landsat cloud score algorithm from the Google Earth Engine to further remove clouds (Gorelick et al., 2017). By incorporating visible, near infrared, shortwave infrared and thermal infrared bands, this algorithm computed cloud score from 0 (clear pixel) to 100 (most likely cloudy pixels) (https://code.earthengine.google.com/dc5611259d9ccab952526b3c2d05ce07).

We further used the Normalized Difference Snow Index (NDSI) to exclude the influence of snow in all images.





$NDSI = \frac{\rho_{swir} - \rho_{grn}}{\rho_{swir} + \rho_{grn}}$ (2)

where $\rho_{swir}$ and $\rho_{grn}$, are shortwave infrared (1550-1750 nm) and green (520-600) bands of Landsat TM/ETM+/OLI images, respectively. From our tests we found NDSI > 0.4 can remove most snow.

### 2.2.2 Ordinary least square (OSL) regression and landslide surface recovery prediction

By nature, vegetation index would change gradually in a year-round (Chen et al., 2004; Yang and Qi, 2017) and the OLS
regression is a commonly used way to restore bad observations by clouds and snow in time series of Landsat images (Zhu and Woodcock, 2014; Zhu et al., 2019). In this work, we performed the OLS for both the pre- and post-seismic EVI time series. We calculated EVI on 15 July for all years, because at this time of the year, the solar incidence angle is near the highest, which could minimize the influence of mountain shadows in rugged terrains. In addition, EVI is near its annual peak values on 15 July, when it is less likely influenced by inter-annual fluctuations of vegetation phenology.

From previous works (Yang and Qi, 2017; Yang et al., 2018b), it is justifiable to assume annual vegetation index on most landslide surfaces change linearly in post-seismic years. For each pixel, the OSL regressed EVI values in all post-seismic years (2008-2019) on 15 July has been used to construct a linear regression line and the slope of the line for each location is used as landslide recovery rate.

### 2.2.3 Analyses of influencing factors

We used the landslide inventory interpreted by Xu et al. (2014) to select Landsat observations on co-seismic landslides. To explain the spatial patterns of landslide recovery, EVI increasing rates were plotted with twelve environmental factors, including: slope, pre-seismic vegetation condition, landslide depth, landslide area, elevation, ground peak acceleration of the earthquake, aspect, slope curvatures, topographic positions, mean annual precipitation, ground cohesion strength and vegetation types.

The landslide area is an attribute of the landslide inventory interpreted by Xu et al. (2014). Ground peak acceleration of the 2008 earthquake can be found from the USGS ShakeMap. Gallen et al. (2015) derived near-surface cohesion by incorporating fracturing. Their derived rock strength was used to study its influence on landslide surface recovery in this work. Mean annual precipitation was provided by the Institute of Mountain Hazards and Environment, CAS (http://english.imde.cas.cn/). Vegetation type data is from Zhang et al. (2007) and this data has also been used in other related works (Yang and Qi, 2017).

All other topographic factors were derived from a 25 m DEM digitized from a 1:50,000 topography map before the earthquake (also used by Fan et al., 2012). Percent slope, aspect directions, planform and profile curvatures were calculated from the DEM in ArcMap 10. Aspect was transformed into aspect index $I_{asp}$ by the following formula:

$$I_{asp} = |COS((aspect - 22.5)/180 \times \pi/2)| \qquad (3)$$

where, aspect is calculated from ArcMap 10. $I_{asp}$ ranges from 0 to 1. Southern aspects have lower values near 0, whereas
northern aspects have higher values near 1.



In our study area, we randomly generated one million points. There are 47,076 points fall within co-seismic landslides. We picked out all abovementioned factors from these points on co-seismic landslides and studied their relation with landslide surface recovery.

## 3 Results

### 145    3.1 Spatial patterns of landslide recovery

Fig. 2 shows landslide surface changing rates measured in annual EVI increasing rate over the entire study area. Annual landslide recovery rates are shown in four categories: $<7\times10^{-3}$, $7$-$14\times10^{-3}$, $14$-$21\times10^{-3}$, and $>21\times10^{-3}$. The spatial pattern of landslide recovery in the study area is heterogeneous. Most landslides near the epicentre area, Yingxiu, have the best recover rate (blue pixels) during the study period, whereas landslides (red pixels) on high mountains (Fig. 2a) are poorly recovered. 150 The poorest recovery of landslide surface is found located on top of the Longmen mountain. These landslides are probably still bare debris with little vegetation cover.

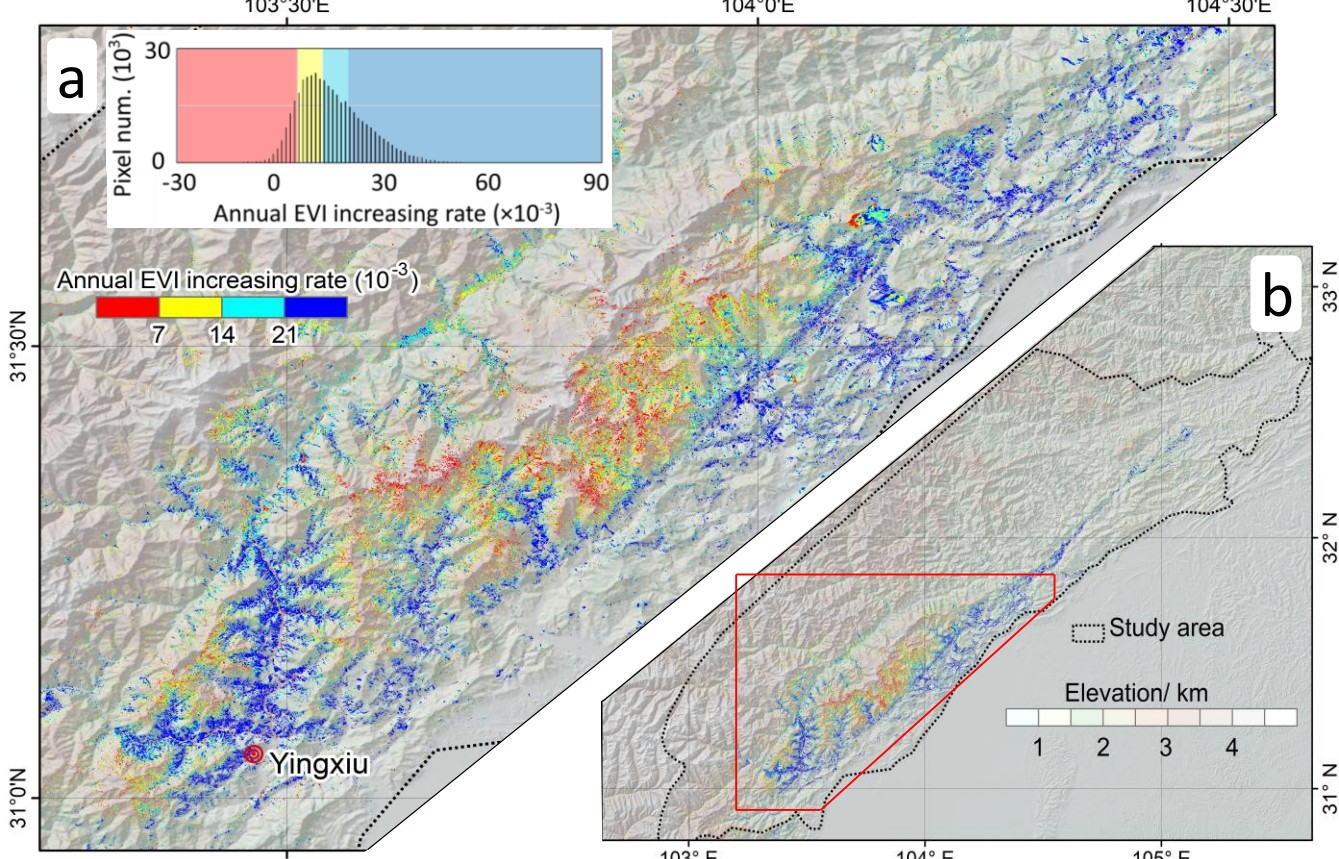

**Figure 2: A heterogeneous spatial pattern of the co-seismic landslide surface recovery after the Wenchuan earthquake. The shown DEM is SRTM DEM downloaded from the USGS website. The hillshade data is produced by the authors by using ArcGIS.**



Inset in Fig. 2 is a histogram showing the frequency distribution of annual EVI increasing rate on co-seismic landslides. Landslide recovery rate ranges from -30 to 90 ×$10^{-3}$, but more than 99% landslide surfaces have positive rates. The mean recover rate on all co-seismic landslides in our study area is 17×$10^{-3}$. 11.4% co-seismic landslides have an annual EVI increasing rate <7×$10^{-3}$ and the rate on 31.4% co-seismic landslides ranges are within 7-14×$10^{-3}$. Annual EVI increasing rate on the other two categories, 14-21×$10^{-3}$ and 21×$10^{-3}$, cover 27.2% and 30.0% of co-seismic landslides in the study area (inset

in Fig. 2a), respectively.

**3.2 Relations between landslide recovery and environmental factors**

To decipher possible influencing factors on landslide surface recovery, we analysed the relations between annual EVI increasing rate and twelve environmental factors: elevation, slope, aspect, topographic position, planform curvature, profile curvature, mean annual precipitation, ground peak acceleration, landslide area, pre-seismic mean EVI, near surface effective

cohesion and vegetation type (Fig. 3). Although annual EVI increasing rates have a wide range (-30-90×$10^{-3}$), 82.0% co-seismic landslides have recovery rates between 0 to 25×$10^{-3}$. In Fig. 3, we can see that these twelve factors influence landslide recover rates are within the range of 0 to 25×$10^{-3}$.

To clearly compare the influence of different factors on annual EVI increasing rate, all figures in Fig. 3 have the same scale of vertical axis (i.e. the annual EVI increasing rate ranges from 0 to 25×$10^{-3}$). Among these twelve factors, elevation, TPI and

pre-seismic EVI are the top three factors that have a wide influencing range. Elevation and TPI have similar negative relations with landslide recover rates. Recover rate decreases when elevation or TPI have higher values. Above an elevation of 1400 m, EVI increasing rate decrease 0.7×$10^{-3}$ per 100 elevation increase. EVI increasing rate increases with the multi-year mean pre-seismic EVI, when the pre-seismic value is lower than 0.75.

In general, ground peak acceleration and landslide area have complex positive relations with the annual EVI increasing rate.

Aspect index and near surface effective cohesion have slight negative relations with the annual EVI rate. The other factors, such as the mean annual precipitation, slope, curvatures and vegetation type seems to have little influence on the annual EVI increasing rate.



**Figure. 3. Relations between environmental factors and landslide recovery rate (measured as annual EVI increasing rate). Lines in a-l are mean values and grey zones are standard error of the means. Point and bars in k and l are mean values and SEM, respectively.**



## 4 Discussion

### 4.1 Heterogeneous recovery of landslide surface after the Wenchuan earthquake

Monitoring post-seismic landslide changes needs large coverage and long-term observation efforts, because earthquake-triggered landslides are widely distributed (Xu et al., 2014) and are very dynamic in post-seismic years (Li et al., 2018; Yang, et al., 2017). By using high spatial resolution images, previous studies found significant and consistent recovery of co-seismic landslide surface and predicted that landslide surface can be fully recovered by vegetation in around a decade in partial region (near the epicentre area and the Mianyuanhe area) (Fan et al., 2018; Li et al., 2016; Tang et al., 2016; Yang et al., 2017). Extrapolations of observations from part of the earthquake-affected area to the whole region may not hold true. For the entire earthquake-affected region, we studied post-seismic vegetation recovery on landslides and found the recovery of co-seismic landslides is spatially heterogeneous. Co-seismic landslides in high Longmen mountain have very poor recovery performances. Results of this work support the findings made by Yang and Qi (2017), but it has much higher spatial resolution to reveal more detailed spatial patterns. Our results suggest distinct patterns of landslide evolution may exist in different parts of the earthquake-affected region in post-seismic years.

### 4.2 Influencing factors on post-seismic landslide recovery

To explain the spatially heterogeneous pattern of landslide recovery, we used twelve environmental variables. For the entire region, elevation, TPI and pre-seismic EVI have the most significant influences. The negative relation with TPI indicate that landslides recover faster on valleys (lower than its nearby locations) than on ridges (higher than its nearby locations). This finding indicates that landslide deposits may stay on hillslopes rather than flow down to valleys in post-seismic years (Fan et al., 2018). Although we cannot quantify the volume of landslide deposits that stayed on hillslopes, a large part of these materials cannot be transported, which may influence the competition between co-seismic uplift and erosion in this region (Parker et al., 2011).

Its relation with elevation indicates that higher elevation has lower temperatures and may lead to slower recovery rates. The pre-seismic EVI is an important indicator for plant physiology and its positive relation with the landslide recover indicates earthquake-triggered landslide did not significantly change vegetation growing conditions. These three dominant factors indicate that post-seismic landslide recovery is collaterally controlled by topography and plant physiology conditions. Although not all factors pose the same significant influences on landslide recovery, different ones may dominate the recovery process at the local scale.

The relation with mean annual precipitation is complex. It is known that climate drives erosion of landslide deposits (Tolorza et al., 2019; Zhang et al. 2019). Precipitation also plays a positive role in vegetation recovery, which is against erosion processes on landslide surfaces (Yang et al., 2018b). To quantify the role of climate on landslide erosion, detailed information on rainfall events, such as duration and intensity should be known. The effects of an intense rainfall within a few minutes is





different to the same amount of precipitation that last many days. The relation between climate and post-seismic erosion may be re-assessed by considering vegetation dynamics.

## 4.3 Implication of landslide surface recovery

Destruction of vegetation by co-seismic landslides leave unstable slope surface exposed and vegetation recovery on landslides could improve shallow slope stability (Kim et al., 2017). The recovery of landslide surface can be used to indicate the duration of a major mountain earthquake (Yang et al., 2018a). Root systems of recovered vegetation could improve soil strength preventing further slope mobilization (Shiels and Walker, 2013) and its canopy can mitigate rain drop splash on landslides, mitigating surface erosion. In addition, recovered vegetation can change landslide surface hydrology in three ways: 1) canopy
intercept rainfall and decrease surface flow, 2) root system improve preferential flow, both can postpone soil saturation and prevent further landsliding; 3) evapotranspiration drain soil water, further decreasing soil moisture (Cowpertwait and Metcalfe, 2009; Meusburger et al., 2010). Despite the root system of recovered vegetation may only influence shallow landslides, its hydrological influence may have a deeper impact on substrates and bedrock weathering beneath top soils, which may relate to deep-seated landslides.

This work only studied landslide surface by using EVI as an index, yet recovery of landslide interior structure still deserves further examination. Although recovered vegetation is fragile and very sensitive to post-seismic mass wasting processes, its recolonization marks the most significant feature changes in post-seismic mountains and the recolonization of vegetation on landslide indicates an at least transient stability of the surface. Vegetation recovery on co-seismic landslide surface could also plays a positive role in reducing surface erosion.

## 4.4 Advantages of Google Earth Engine

Major earthquakes, such as the Wenchuan earthquake, could trigger numerous landslides spreading very large regions. Studying regional landslide changes after major earthquakes is important for understanding geo-hazards after major disturbances, yet it is technically difficult to implement. This is because mapping landslides over very large regions requires many optical images of high quality. These images should not be covered by clouds, and should be acquired in summer seasons,
when landslides are easily recognized from surrounding ground features. Images that meet these criteria is rare, especially in monsoon climate regions. Therefore, most works used a sub-set of the earthquake-affected region to study landslide changes. This is especially true for the Wenchuan region (Fan et al., 2018; Yang et al., 2015; 2017).

This work used a cloud-based platform, GEE, to map landslides using all available Landsat images from 2001 to 2019 over a large region. GEE is a cloud-based platform, on which users can write simple scripts on their personal computers and perform
heavy computations on numerous Google's infrastructures. Using this platform, we can easily find replacement pixels from nearby date images to replace cloud contaminated pixels. This work only used Landsat images, because this is a consistent data with a long archive. Besides the Landsat data, other datasets such as the Sentinel-2 data are also available (Yang et al., 2019).



In addition, the GEE can provide timely data to anywhere on the planet, which is crucial for hazard studies. The GEE updates
its datasets at a daily level. This is important for disaster response. For example, after the Jinsha river landslide and Yarlung
Zangbo debris flow, impacts of these catastrophic geomorphic processes can be quickly assessed (Yang et al., 2019). By
combining machine learning with the cloud-based platform, it has the potential to utilize large volumes of remote sensing data
efficiently and has the potential to provide a new way in fast hazard mappings.

## 5 Conclusions

The 2008 Wenchuan earthquake triggered more than 190,000 co-seismic landslides spreading many thousand square
kilometres. Mapping landslide recovery on these co-seismic landslides could be difficult. Using a cloud-based computing
platform, GEE, this work mapped surface recovery of all co-seismic landslides with Landsat images from 2001 to 2019. We
found >99% landslide surfaces have been recovering since 2008, but the spatial pattern is very heterogeneous. In general, co-
seismic landslides on higher elevation ridges recover slower than landslides on valleys with lower elevations. Landslides on
the high Longmen mountain have the slowest recover rates, whereas the epicentre area along the Min river valley has the
fastest rates. Elevation, TPI and pre-seismic vegetation condition are top three most important factors that influence post-
seismic landslide recovery. In addition, pre-seismic EVI also have a strong relation with landslide recovery. Locations with
better vegetation growing conditions before the earthquake usually have faster landslide recover rates.

## Acknowledgements

This work was jointly supported by the National Science Foundation of China (NO. 41807500) and the "Fundamental Research
Funds for the Central Universities" (No. 2019ZY33). The authors would like to show their gratitude to Professor Jing Liu-
Zeng from China Earthquake Administration for constructive discussions and Dr. Chao Ma to provide the mean annual
precipitation data. The GEE code to derive recovery rate of coseismic landslides can be accessed at:
https://code.earthengine.google.com/a489a5beb38e7d9f81d832306cf92e46.

## Data availability


Ground peak acceleration of the earthquake is available from the USGS website. Ground cohesion strength is provided by
Sean F Gallen (Sean.Gallen@colostate.edu). Elevation, slope, slope curvatures, aspect and topographic positions are derived
from a 1:50,000 topographic map, which is provided by Prof. Cees van Westen (c.j.vanwesten@utwente.nl). The landslide
inventory of the 2008 Wenchuan earthquake is provided by Prof. Chong Xu (xc11111111@126.com). Mean annual
precipitation data is provided by Dr. Chao Ma (sanguoxumei@163.com).



## Author contribution

W.Y., W.Q. and J.F. conceived the work. W.Y. and W.Q. performed the analysis in GEE, J.F. analysed the factor analysis part.

## Conflict of interest

The authors have no conflict of interest.

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
