# Peer review of "Using Google Earth Engine to monitor co-seismic landslide recovery after the 2008 Wenchuan earthquake"

_Earth Surface Dynamics, 2020_

## Referee Comment (RC1) · Odin Marc (Referee) · 20 Jan 2021

The authors present a regional analysis of remotely sensed vegetation index in the Wenchuan region to assess the dynamics of landslide surface revegetation and how it depends on various environmental controls. Many studies (including several by the same authors have done similar exercise in the last 5 years (eg, Yang et al 2017, 2018a, 2018b, Yunus et al 2020), the novelty of this work is that it covers an area larger than previous studies, that it is based on Landsat imagery (with a finer resolution 30m than previous work based on Modis) and done with Google Earth Engine. The topic could be of interest to Esurf I would say and the structure and writing of the manuscript

is fairly good. However, the main problem is that beyond the technical novelty stated above the manuscript does not bring much and appears to bring very little new scientific content (Beyond the site there are only 2 figures, which basically are not very different from what can be find in Yunus et al 2020 or previous paper of the author Yang et al 2018a,b).

So it may be better redirected to a regional journal ? Or become a technical note in Esurf ? But even if it is retargeted like this it would still require substantial modification. Indeed the methodology is not always sufficiently described, and the discussion is often quite poor. I give below the major limits of the work and a series of Line by Line recommendation.

Odin Marc

Major comments:

1) The author use the term of "landslide (surface) recovery" which is inaccurate and too vague relative to their methods. Recovery can mean many things, while the author are very specifically measuring the temporal evolution of vegetation index. So I think they should replace everywhere by "Landslide (surface) revegetalisation" . Below I gave a few of the lines where this occur but the change should be through out unless the authors have sentence where they specifically discuss various option for landslide recovery (deposit armoring , or other form of grain size changes; Soil formation, bedrock/regolith properties changes ?)

2) I think the Pre-seismic EVI should not be used as a controlling factor but used to normalize the EVI increase rate. The simplest scenario of revegetalisation is that vegetation similar to the pre-seismic vegetation will recolonize the landslide areas. Thus a large pre-seismic EVI will tend to a large post seismic EVI and vice versa for small pre-seismic EVI. As a result it is intuitively expected that the absolute recovery rate is correlated with pre-seismic EVI. I think it may be more interesting to present all the result in terms of EVI/pre-seismic EVI to obtain a score from 0 to ~100% which would

mean a full recovery. The rate of this normalized revegetalisation could be analyzed relative to all factors as done by the authors.

3) The result analysis is clearly lacking a cross-correlation analysis... Why to study Elevation and TPI if they both are very highly correlated ? Elevation is the physically meaningful one as it related to temperature and thus the vegetation type and probably growth rate ... TPI is less meaningful for a plant I suspect... Also I suspect Pre-seismic EVI and Elevation to be strongly correlated (because of temperature...), hence the importance to study EVI/EVI_pre-seismic (see comment 2). Actually Fig 4 of Yang et al 2018b is exactly showing that PreEQ-NDVI and Elevation are strongly related. So it makes no sense to treat Elevation and Pre seismic EVI as independent, and strongly supports the normalization by pre-EQ EVI.

4) The methods miss many details : How were treated pixel relative to landslide boundary (i.e. when they were partly across a slide and partly across undisturbed slope ?). What about seasonal variations ? Several previous studies (eg, Yang et al 2018a, Fig 3) do account for their treatment allowing to have a finer control on the revegetalisation. I wonder why this is not done here.

Last, the author state they have shown that Linear recovery is the best assumption, which leaves me very skeptical. Even in their own work ( Yang et al 2018a, Fig 3) an example of recovery of the NDVI is shown and it is quite clear that the revegetalisation is far from being linear : there is a few years (3-4) with no NDVI trend , then an increase in NDVI which seems to accelerate. This could make sense, as there may be some delay before new plant colonize the area, and then as vegetation develop multiple species can make the rate of biomass accumulation (and NDVI increase) increase with time, giving a non linear revegetation rate. One way to make the work scientifically richer could be to specifically extract various parameters describing the shape of the revegetalisation curve (EVI or NDVI trend) for as many landslide pixel as possible... Some other aspects are poorly detailed (see Line by Line comments below).

5) After accounting for the changes in the other major points, the discussion should be substantially improved, and I suggest the author to also compare their work and result to other studies which have looked at revegetation on landslides not necessarily only in the Wenchuan area.

Two Ex from Taiwan (other exist) Schomakers, J., Jien, S.-H., Lee, T.-Y., Huang, J.-C., Hseu, Z.-Y., Lin, Z. L., Lee, L.-C., Hein, T., Mentler, A. and Zehetner, F.: Soil and biomass carbon re-accumulation after landslide disturbances, Geomorphology, 288(Supplement C), 164–174, https://doi.org/10.1016/j.geomorph.2017.03.032, 2017. Lin, W. T., Lin, C. Y. and Chou, W. C.: Assessment of vegetation recovery and soil erosion at landslides caused by a catastrophic earthquake: a case study in Central Taiwan, Ecological engineering, 28(1), 79–89, 2006.

Line by Line comments L62 : No need to say its a cloud based: "In this work we use the GEE". ALso surface recovery is ambiguous. I would say : "to track the revegetalisation of coseismic landslide" rather than map surface recovery...

L64: "is easy to use" rather than has the advantage of easy-to-use ... L64-65: switch the subject : "allows researcher unfamiliar with remote sensing techniques to process efficiently large number of images "

L65, 89, 98 , 115 , 132, 143, 161 etc etc: replace recovery by revegetalisation.

L84: What do you mean the best processing level ? Clarify or remove.

L113, 121 : replace OSL by OLS

L114: replace "in a year-round" by seasonally

L117-119 : I do not understand why the authors estimate only the 15 of July vegetation level... The vegetation EVI (as well as solar radiation in relation to view angle/passage time) must vary in a seasonal repeatable way. So it should be possible to stack the EVI of every months preceding the EQ to obtain an annual cycle of EVI and be able to compare this to the months and year following the EQ... It would allow to have a much

finer temporal resolution of the vegetation recovery.

L127: remove "including" it is unecessary and rather confusing.

Paragraph 2.2.3 : Should be written "Peak Ground Acceleration" everywhere.

L131 : Description of Gallen 2015 data is inaccurate and incomplete. The authors must improve that.

L134 : What contains ZHang 2007 ? Vegetation type at which resolution ? With which value ? This need to be clarified and possibly example data be shown in the supplement...

L141: I guess you mean "we randomly selected 1 million pixels" . If not please clarify.

L156 : You should say here something like " Therefore, we exclude the negative EVI changes"

L157: The "mean revegetalisation rate"

L162-165 : This is a repetition from the methods , to be removed...

L165-166 : Repetition from the result sentence to remove or change.

L166-167 : Do not understand this sentence. To rephrase.

L169 : You did not define TPI. Guess it is Topographic Position Index (to be defined/reference in the method)

165-177 : This paragraph has poor writing ... and is insufficiently quantitative : Could we have the correlation coefficient ?

Fig 3 : How was landslide depth obtained ? This should be detailed in the method section... Or depth should be removed (possibly better if it is just a rescaling of Area) Tangent curvature was in the method among the 12 and disappeared from this figure ?

L195-200 : I think all this argument about TPI are unlikely. Instead I think TPI is highly

correlated to ELevation (=Temperature) which is most likely the control.

L200-210 : Temperature role is underplayed. Correlation between Pre-seismic EVI and ELevation should also be assessed, as they very likely are correlated. The role of climate, or landslide area and slope likely cannot be assessed before the normalization by EVI_pre is not done...

L217: "The recovery of landslide surface can be used to indicate the duration of a major mountain earthquake (Yang et al., 2018a)" This sentence makes no sense. To be removed or rephrased.

L228-229 : Last sentence is a repetition of what is above. TO be removed.

Paragraph 4.4 : Rather useless in my opinion. Sure GEE stores imagery and can process lot of data for various purposes, but no need to write 15 lines about it with tons of self citation ...(Just cite Gorelick, as done before... Further the focus of past studies on specific zones is due to the fact they aimed at mapping landslides accurately, which is not what the author are doing. They use maps from other (made often with resolution higher than Landsat) and then extract pixels within them to track the evolution of their EVI.

So L 238 "to map landslides using all available Landsat" is simply a wrong statement to be removed. GLobally this paragraph should be mostly shrinked/removed.

Conclusions : "We found >99% landslide surfaces have been recovering since 2008" I would rather say "surfaces have experienced some revegetalisation as tracked by EVI increased" or something like that, less ambiguous.

---

## Referee Comment (RC2) · Anonymous Referee #2 · 20 Jan 2021

The paper describes the application of Google Earth Engine in mapping the vegetation growth in the earthquake-stricken region in Wenchuan. The authors combine GEE and EVI, and various geo-environmental factors to shed light onto the evolution of landslide stabilization or vegetation recovery in the study region since 2008. I found the approach using GEE intriguing in general and see a benefit in fast processing for such kind of studies. However, at the same time the study suffers from several flaws that challenge a publication in its current shape. At the current state, I see lack of novelty in the overall assessment. Currently, the article generates little incremental knowledge. The workflow has already been presented in Jiang et al., 2015; Yang etal., 2017 and Yunus et al., 2020; except that instead of MODIS, this study used Landsat, and instead of

[Figure]

NDVI, here EVI based method is chosen. Major comments: The workflow and evolution of landslides following Wenchuan earthquake have been described in detail by many studies. So, without describing a new method (other than GEE implementation) and without providing substantial original insight to vegetation regrowth (TPI and Elevation are basically the same), the scope of the study shrinks to the technicality of GEE and a case information. In section 2.2.2 authors described that they calculated EVI on 15th July. Well I agree with the date corresponds to growing season, but it is unclear that how the authors get data for July 15th every year for all the tiles necessary to generate EVI map. This part of methodology is very vague. The limited growth of vegetation on higher elevated region can be because of several reason: persistent snow cover/ no loose materials to grow the roots or bare rocks / active landslides – high erosion / climate (rainfall and temp.). It is important to investigate these areas in detail using google earth images and deepen your discussion section. Figure 2. it is quite misleading by showing high values of annual EVI increasing rate and then understands it is to multiply by $10^{-3}$. Is this increasing rate statistically significant ?

Figure 3 C. How do authors calculate the landslide depth ? (source).

Figure 3 L : basically all the type of trees/plants is damaged after the earthquake . So the comparison shown in Fig 3 L is meaningless

Author wrote "We found >99% landslide surfaces have been recovering since 2008" . Where did this value comes from ?. Section 4.4 is not a proper discussion. I am not qualified to judge the English, but I feel there is considerable scope to improve. For eg., Line 25 – 'thousands of hundreds'

---

## Author Comment (AC1) · 20 Mar 2021

Major comments: 1) The author use the term of "landslide (surface) recovery" which is inaccurate and too vague relative to their methods. Recovery can mean many things, while the author are very specifically measuring the temporal evolution of vegetation index. So I think they should replace everywhere by "Landslide (surface) revegetalisation" . Below I gave a few of the lines where this occur but the change should be throughout unless the authors have sentence where they specifically discuss various option for landslide recovery (deposit armoring, or other form of grain size changes; Soil formation, bedrock/regolith properties changes ?)

Response: We examined every place and replaced the "landslide (surface) recovery" with "landslide revegetation".

2) I think the Pre-seismic EVI should not be used as a controlling factor but used to normalize the EVI increase rate. The simplest scenario of revegetalisation is that vegetation similar to the pre-seismic vegetation will recolonize the landslide areas. Thus a large pre-seismic EVI will tend to a large post seismic EVI and vice versa for small pre-seismic EVI. As a result, it is intuitively expected that the absolute recovery rate is correlated with pre-seismic EVI. I think it may be more interesting to present all the result in terms of EVI/pre-seismic EVI to obtain a score from 0 to âĹij100% which would mean a full recovery. The rate of this normalized revegetalisation could be analyzed relative to all factors as done by the authors.

Response: We agree with you that pre-seismic EVI should not be used in our analysis. Our later cross-correlation analysis also confirmed that pre-seismic EVI is significantly correlated with elevation and is not considered as an influencing factor in later analysis. However, we did not use EVI/pre-seismic EVI. Our reason is based on the logic that if all post-seismic EVI is divided by the same pre-seismic EVI, the changing rate will not change. That means that for each pixel, EVI on the 15th July of post-seismic years divided by the same mean pre-seismic EVI will not change the changing rate. However, we tried to solve this problem by another way. We used our model to calculate the recovery time of the vegetation damaged by landslides to 100% of pre-seismic EVI and added a new result as Fig. 1 in this response letter.

Response Figure 1. Predicted recovery time of the vegetation on landslides for the entire Wechuan earthquake affected region. (Added as Fig. 3 in the revised manuscript.)

3) The result analysis is clearly lacking a cross-correlation analysis... Why to study Elevation and TPI if they both are very highly correlated ? Elevation is the physically meaningful one as it related to temperature and thus the vegetation type and probably growth rate ... TPI is less meaningful for a plant I suspect... Also I suspect Pre-seismic

EVI and Elevation to be strongly correlated (because of temperature...), hence the importance to study EVI/EVI_pre-seismic (see comment 2). Actually Fig 4 of Yang et al 2018b is exactly showing that PreEQ-NDVI and Elevation are strongly related. So it makes no sense to treat Elevation and Pre seismic EVI as independent, and strongly supports the normalization by pre-EQ EVI.

Response: We added an analysis of cross-correlation among the originally used 12 variables (Fig. 2 of this response letter). We deleted TPI and pre-seismic EVI in later analysis because we found elevation is correlated with both variables (R = 0.44 and -0.55, also pointed by the reviewer). The landslide depth is a rescaling of Area. The formula comes from Xu et al. (2016, Scientific Reports, 6, 29797). The correlation coefficient (R) between landslide depth and landslide area is 0.11 and we deleted the depth variable in our later analysis. We found plan curvature and profile curvature are negatively correlated (R = -0.35) in this region and we dropped the profile curvature as this variable also has higher correlations with other variables (such as aspect). We also gave up TPI and pre-seismic EVI in later analysis. After the cross-correlation analysis, the following eight variables are left: landslide area, aspect, elevation, PGA, precipitation, profile curvature, rock strength and slope.

Response Fig. 2. Cross-correlation analysis among all influencing factors. (Added as Fig. 4 in the revised manuscript.)

4)-1 The methods miss many details: How were treated pixel relative to landslide boundary (i.e. when they were partly across a slide and partly across undisturbed slope ?).

Response: Only pixels with their center points in a landslide are determined as a landslide point and used in our later analysis. This procedure will not influence our results, because EVI are continuous variables. Even a pixel covers part of a landslide, its EVI is probably lower than pre-disturbed values.

4)-2 What about seasonal variations? Several previous studies (eg, Yang et al 2018a,

Fig 3) do account for their treatment allowing to have a finer control on the revegetalisation. I wonder why this is not done here.

Response: In the previous work (Yang et al. 2018a, Fig. 3), we used the time series analysis to extract the trend signal from the data. Because we established post-seismic EVI for the study area, we can get EVI on any date in theory. To remove the influence of EVI fluctuations with seasons, we decided to select the 15th July (the same date) of different years to assess vegetation recovery.

4)-3 Last, the author state they have shown that Linear recovery is the best assumption, which leaves me very skeptical. Even in their own work (Yang et al 2018a, Fig 3) an example of recovery of the NDVI is shown and it is quite clear that the revegetalisation is far from being linear : there is a few years (3-4) with no NDVI trend , then an increase in NDVI which seems to accelerate. This could make sense, as there may be some delay before new plant colonize the area, and then as vegetation develop multiple species can make the rate of biomass accumulation (and NDVI increase) increase with time, giving a non linear revegetation rate. One way to make the work scientifically richer could be to specifically extract various parameters describing the shape of the revegetalisation curve (EVI or NDVI trend) for as many landslide pixel as possible... Some other aspects are poorly detailed (see Line by Line comments below).

Response: We acknowledge that post-seismic EVI may not recover linearly in the longer term as mentioned by Schomakers et al. (2017), which studied 40 years of vegetation recovery on landslides. In this work, we only used EVI values in a few post-seismic years (2008-2019). As the sample for post-seismic EVI is too few (12 years or EVI values each pixel) that it may be more suitable to use the simplest linear model to fit for the limited observations. In addition, as our model to fit for the EVI time series is a linear one, its derived EVI values are also changing linearly. In our future work, we may try to extract raw EVI values with non-linear models when more observations are available.

5) After accounting for the changes in the other major points, the discussion should be substantially improved, and I suggest the author to also compare their work and result to other studies which have looked at revegetation on landslides not necessarily only in the Wenchuan area. Two Ex from Taiwan (other exist) Schomakers, J., Jien, S.-H., Lee, T.-Y., Huang, J.- C., Hseu, Z.-Y., Lin, Z. L., Lee, L.-C., Hein, T., Mentler, A. and Zehetner, F.: Soil and biomass carbon re-accumulation after landslide disturbances, Geomorphology, 288(Supplement C), 164–174, https://doi.org/10.1016/j.geomorph.2017.03.032, 2017. Lin, W. T., Lin, C. Y. and Chou, W. C.: Assessment of vegetation recovery and soil erosion at landslides caused by a catastrophic earthquake: a case study in Central Taiwan, Ecological engineering, 28(1), 79–89, 2006.

Response: Thanks for this comment, the discussion part has been changed by comparing this work with others, including related works done in Taiwan:

"There are many early works in Taiwan that monitor vegetation recovery on landslides (Lin et al., 2004; Lin et al., 2005; Lin et al., 2008b). But the spatial extent of their study area are very limited. There are also very few long-term examinations of vegetation recovery after the 1999 Taiwan earthquake in recent years. For the Wenchuan region, there has been consistent efforts to monitor vegetation recovery in post-seismic years."(in sub-section 4.1)

Other major changes were made at the beginning (prior to sub-section 4.1) and end of the original discussion:

Two new paragraphs in the beginning of the discussion part.

"Major earthquakes, such as the Wenchuan earthquake, could trigger numerous landslides spreading very large regions (Xu et al., 2014). Studying post-seismic landslide changes is important for understanding geo-hazard evolution (Fan et al., 2018; Marc et al., 2015; Tang et al., 2016). To interpret post-seismic landslides, remote sensing images should meet the following criteria: 1) cloud-free images is a basic requirement; 2) to overcome the influence of mountain shadows in complex terrains, images acquired at high solar altitude angles are favoured; 3) images acquired in summer seasons are favoured to have large spectral contrast between landslides and background vegetation. It is often challenging to acquire remote sensing images that meet these criteria for the entire earthquake-affected region. In addition, active post-seismic landslides are more difficult than fresh ones to interpret because the contrast between active landslides and the background stable slopes is less obvious while vegetation are recovering. The transition from active post-seismic landslides to a steady slope is a continuous process but the interpretation of post-seismic landslides is a binary way with subjectivity.

To avoid abovementioned problems, vegetation dynamics on landslides' surface provides a substitute index for studying post-seismic landslide changes and have been intensively studied (Jiang et al., 2015; Lin et al., 2004; Yang et al., 2018a; Yunus et al., 2020). However, cloud contamination remains a challenge for monitoring vegetation recovery for a large region. Previous works either use coarse remote sensing images (such as MODIS) (Yang et al.,2018a) or studied part of the earthquake-affected region (Li et al., 2016) because monitoring the entire region requires many optical images of high quality. In this work, we eliminated cloud-contaminated pixels by using the cloud mask and used a linear model to interpolate those masked EVI values for each pixel. By using this method, we are able to monitor continuous vegetation recovery for the entire earthquake-affected region." The original sub-section 4.4 was removed.

Line by Line comments

1. L62 : No need to say its a cloud based: "In this work we use the GEE".

2. ALso surface recovery is ambiguous. I would say : "to track the revegetalisation of coseismic landslide" rather than map surface recovery...

3. L64: "is easy to use" rather than has the advantage of easy-to-use ...

4. L64-65: switch the subject : "allows researcher unfamiliar with remote sensing techniques to process efficiently large number of images "

5. L65, 89, 98 , 115 , 132, 143, 161 etc etc: replace recovery by revegetalisation.

6. L113, 121 : replace OSL by OLS

7. L114: replace "in a year-round" by seasonally

8. L127: remove "including" it is unecessary and rather confusing.

We agree with comments 1-8 and solved all these raised issues.

9. L84: What do you mean the best processing level ? Clarify or remove.

Response: The sentence is removed.

10. L117-119 : I do not understand why the authors estimate only the 15 of July vegetation level... The vegetation EVI (as well as solar radiation in relation to view angle/passage time) must vary in a seasonal repeatable way. So it should be possible to stack the EVI of every months preceding the EQ to obtain an annual cycle of EVI and be able to compare this to the months and year following the EQ... It would allow to have a much finer temporal resolution of the vegetation recovery.

Response: In theory, we can derive any data from the OLS regressed time series of vegetation index. To detect vegetation recovery, we only need to get the changing trend from the time series data. Because the chosen model is a linear one, the trend derived from any date are the same as long as it is the same specific day in different years. We chose the 15 July of each year because "at this time of the year, the solar incidence angle is near the highest, which could minimize the influence of mountain shadows in rugged terrains. In addition, EVI is near its annual peak values on 15 July, when it is less likely influenced by inter-annual fluctuations of vegetation phenology". We added a few sentences to explain the reason to only select the 15 of July.

11. Paragraph 2.2.3 : Should be written "Peak Ground Acceleration" everywhere.

Response: We changed the first one and used the PGA elsewhere in the main text.

12. L131 : Description of Gallen 2015 data is inaccurate and incomplete. The authors must improve that.

Response: The original sentence is not a description of the data from Gallen et al. (2015). It intends to describe how to use the data. To make it clearer, the original sentence has been changed from

"Gallen et al. (2015) derived near-surface cohesion by incorporating fracturing. Their derived rock strength was used to study its influence on landslide surface recovery in this work."

to

"Gallen et al. (2015) estimated near-surface rock strength by considering rock fracturing. In this work, their derived rock strength data was used to study its influence on vegetation recovery at landslides' surface."

13. L134 : What contains ZHang 2007 ? Vegetation type at which resolution ? With which value ? This need to be clarified and possibly example data be shown in the supplement...

Response: The vegetation type is grouped as a few main categories, such as needle-leaf forest, broadleaf forest, shrubland, et al. The map is at a scale of 1:1,000,000. We transformed the map into a spatial resolution of 30m.

14. L141: I guess you mean "we randomly selected 1 million pixels" . If not please clarify.

Response: We changed to "we randomly selected 1 million pixels".

15. L156 : You should say here something like " Therefore, we exclude the negative EVI changes"

Response: We added the sentence.

16. L157: The "mean revegetalisation rate"

17. L162-165 : This is a repetition from the methods , to be removed...

18. L165-166 : Repetition from the result sentence to remove or change.

We agree with comments 16-18 and solved all these raised issues.

19. L166-167 : Do not understand this sentence. To rephrase.

Response: This part has been removed.

20. L169 : You did not define TPI. Guess it is Topographic Position Index (to be defined/reference in the method)

Response: we defined TPI at the first place it appears in the manuscript.

21. 165-177 : This paragraph has poor writing ... and is insufficiently quantitative : Could we have the correlation coefficient ?

Response: We used the Random Forests to rank the importance of these features. The original Lines 165-177 (sub-section 3.2 Relations between landslide recovery and environmental factors) has been changed to:

"To remove dependencies among different all candidate factors, we calculated correlation coefficients among all variables (Fig. 4). We found elevation is highly correlated with TPI (R = 0.44) and pre-seismic EVI (R = -0.55). Elevation is a physically meaningful factor as it relates to temperature and the vegetation type. So we removed TPI and pre-seismic EVI in our later analysis.

The correlation coefficient (R) between landslide depth and landslide area is 0.11. As the landslide depth is a rescaling of Area (Xu et al., 2016), we removed the landslide depth in later analysis. Plan curvature and profile curvature are also negatively correlated (R = -0.35) in this region and we removed the profile curvature because this

variable has higher correlations with aspect. After the cross-correlation analysis, the following eight variables are left: landslide area, aspect, elevation, PGA, precipitation, profile curvature, rock strength and slope. We further used the Random Forests to rank their relative importance on determining post-seismic vegetation recovery (Fig. 5). Our Random Forests analysis indicate that elevation, area and slope are top three most important factors on post-seismic vegetation recovery on landslides, which can explain >57% of all eight considered factors."

Response Fig. 3. Feature importance calculated by the Random Forests. (Added as Fig. 5 in the revised manuscript.)

22. Fig 3 : How was landslide depth obtained ? This should be detailed in the method section... Or depth should be removed (possibly better if it is just a rescaling of Area) Tangent curvature was in the method among the 12 and disappeared from this figure ?

Response: Landslide depth is a rescaling of area. The function is from Xu et al. (2016, Scientific Reports, 6, 29797). After cross-correlation analysis, this variable is dropped out from the Random Forests. The plan curvature is not considered in the revised manuscript.

23. L195-200 : I think all this argument about TPI are unlikely. Instead I think TPI is highly correlated to ELevation (=Temperature) which is most likely the control.

Response: TPI is correlated to elevation (R = 0.44). This argument is removed.

24. L200-210 : Temperature role is underplayed. Correlation between Pre-seismic EVI and ELevation should also be assessed, as they very likely are correlated. The role of climate, or landslide area and slope likely cannot be assessed before the normalization by EVI_pre is not done...

Response: We found pre-seismic EVI is correlated with elevation (R=-0.55) and dropped out pre-seismic EVI in our later analysis. Our Random Forests analysis indicate that elevation, area and slope are top three most important factors on post-seismic

vegetation recovery on landslides.

25. L217: "The recovery of landslide surface can be used to indicate the duration of a major mountain earthquake (Yang et al., 2018a)" This sentence makes no sense. To be removed or rephrased.

Response: Removed.

26. L228-229 : Last sentence is a repetition of what is above. TO be removed.

Response: The sentence is removed.

27. Paragraph 4.4 : Rather useless in my opinion. Sure GEE stores imagery and can process lot of data for various purposes, but no need to write 15 lines about it with tons of self citation ...(Just cite Gorelick, as done before... Further the focus of past studies on specific zones is due to the fact they aimed at mapping landslides accurately, which is not what the author are doing. They use maps from other (made often with resolution higher than Landsat) and then extract pixels within them to track the evolution of their EVI. So L 238 "to map landslides using all available Landsat" is simply a wrong statement to be removed. GLobally this paragraph should be mostly shrinked/removed.

Response: The point we want to make is that monitoring vegetation dynamics on landslides in large region is difficult for the following reasons:

1) images are often contaminated by clouds, 2) remote sensing images acquired in summer seasons is favored, 3) images should be acquired at the same date of different years to eliminate the influence of phenology.

As the authors did lots of imagery search in east Tibetan. For example, Yang (2020, Sensors, 20, 4721) investigated 402 Sentinel-2 images on a slope taken from 2015 to 2019 and found 65.7% of images cannot be used due to cloud contamination. If consider for a larger spatial extent, there would be much more images contaminated. In addition, majority of the clear images are winter seasons and cannot be used to

retrieve EVI or interpret landslides due to vegetation withering, snow cover and heavy mountain shadows. We insist to convey the challenge of using optical images to monitor the dynamics of landslides and vegetation recovery. The original sub-section 4.4 is deleted. We added two new paragraph in the beginning of the discussion section prior to sub-section 4.1:

"Major earthquakes, such as the Wenchuan earthquake, could trigger numerous landslides spreading very large regions (Xu et al., 2014). Studying post-seismic landslide changes is important for understanding geo-hazard evolution (Fan et al., 2018; Marc et al., 2015; Tang et al., 2016). To interpret post-seismic landslides, remote sensing images should meet the following criteria: 1) cloud-free images is a basic requirement; 2) to overcome the influence of mountain shadows in complex terrains, images acquired at high solar altitude angles are favoured; 3) images acquired in summer seasons are favoured to have large spectral contrast between landslides and background vegetation. It is often challenging to acquire remote sensing images that meet these criteria for the entire earthquake-affected region. In addition, active post-seismic landslides are more difficult than fresh ones to interpret because the contrast between active landslides and the background stable slopes is less obvious while vegetation are recovering. The transition from active post-seismic landslides to a steady slope is a continuous process but the interpretation of post-seismic landslides is a binary way with subjectivity.

To avoid abovementioned problems, vegetation dynamics on landslides' surface provides a substitute index for studying post-seismic landslide changes and have been intensively studied (Jiang et al., 2015; Lin et al., 2004; Yang et al., 2018a; Yunus et al., 2020). However, cloud contamination remains a challenge for monitoring vegetation recovery for a large region. Previous works either use coarse remote sensing images (such as MODIS) (Yang et al.,2018a) or studied part of the earthquake-affected region (Li et al., 2016) because monitoring the entire region requires many optical images of high quality. In this work, we eliminated cloud-contaminated pixels by using the cloud

mask and used a linear model to interpolate those masked EVI values for each pixel. By using this method, we are able to monitor continuous vegetation recovery for the entire earthquake-affected region."

28. Conclusions : "We found >99% landslide surfaces have been recovering since 2008" I would rather say "surfaces have experienced some revegetalisation as tracked by EVI increased" or something like that, less ambiguous.

Response: Changed to "We found >99% landslide surfaces have experienced some revegetation as tracked by EVI".
* * *
**Fig. 1.** Predicted recovery time of the vegetation on landslides for the entire Wechuan earth-quake affected region. (Added as Fig. 3 in the revised manuscript.)

| | Area | Plan curvature | Aspect | Elevation | PGA | Precipitation | Pre-seismic EVI | Profile curvature | Depth | Rock strength | Slope | TPI |
|---|---|---|---|---|---|---|---|---|---|---|---|---|
| Area | 1 | <0.01 | 0.02 | -0.02 | <0.01 | 0.11 | 0.05 | <0.01 | 0.11 | -0.08 | -0.01 | 0.02 |
| Plan curvature | <0.01 | 1 | -0.22 | 0.04 | <0.01 | -0.01 | 0.10 | -0.35 | <0.01 | 0.01 | 0.04 | <0.01 |
| Aspect | 0.02 | -0.22 | 1 | -0.02 | <0.01 | -0.03 | 0.03 | 0.16 | 0.04 | -0.01 | -0.06 | -0.13 |
| Elevation | -0.02 | 0.04 | -0.02 | 1 | 0.01 | 0.08 | -0.55 | -0.13 | -0.08 | 0.29 | 0.23 | 0.44 |
| PGA | <0.01 | <0.01 | <0.01 | 0.01 | 1 | -0.01 | -0.01 | <0.01 | <0.01 | <0.01 | <0.01 | <0.01 |
| Precipitation | 0.11 | -0.01 | -0.03 | 0.08 | -0.01 | 1 | <0.01 | -0.01 | 0.09 | -0.02 | 0.04 | 0.13 |
| Pre-seismic EVI | 0.05 | 0.10 | 0.03 | -0.55 | -0.01 | <0.01 | 1 | -0.11 | 0.04 | -0.16 | -0.15 | -0.20 |
| Profile curvature | <0.01 | -0.35 | 0.16 | -0.13 | <0.01 | -0.01 | -0.11 | 1 | 0.02 | <0.01 | -0.06 | -0.18 |
| Depth | 0.11 | <0.01 | 0.04 | -0.08 | <0.01 | 0.09 | 0.04 | 0.02 | 1 | -0.10 | 0.09 | -0.10 |
| Rock strength | -0.08 | 0.01 | -0.01 | 0.29 | <0.01 | -0.02 | -0.16 | <0.01 | -0.10 | 1 | 0.14 | -0.08 |
| Slope | -0.01 | 0.04 | -0.06 | 0.23 | <0.01 | 0.04 | -0.15 | -0.06 | 0.09 | 0.14 | 1 | -0.05 |
| TPI | 0.02 | <0.01 | -0.13 | 0.44 | <0.01 | 0.13 | -0.20 | -0.18 | -0.10 | -0.08 | -0.05 | 1 |

**Fig. 2.** Cross-correlation analysis among all influencing factors. (Added as Fig. 4 in the revised manuscript.)

[Figure]

**Fig. 3.** Feature importance calculated by the Random Forests. (Added as Fig. 5 in the revised manuscript.)

---

## Author Comment (AC2) · 20 Mar 2021

(1) The workflow and evolution of landslides following Wenchuan earthquake have been described in detail by many studies. So, without describing a new method (other than GEE implementation) and without providing substantial original insight to vegetation regrowth (TPI and Elevation are basically the same), the scope of the study shrinks to the technicality of GEE and a case information.

Response: In this work we proposed a new method to generate cloud free images to assess post-seismic vegetation recovery for the entire region. We also used the crosscorrelation and Random Forests to quantify the importance of 12 major influencing factors. Please refer the response to reviewer #1.

(2) In section 2.2.2 authors described that they calculated EVI on 15th July. Well I agree with the date corresponds to growing season, but it is unclear that how the authors get data for July 15th every year for all the tiles necessary to generate EVI map. This part of methodology is very vague.

Response: Thanks for this concern. Because we established a post-seismic EVI model by using EVI observations for every pixel, we can use the model to calculate EVI on the 15th July of every year. We added the model (Eq. 1, please refer to Fig. 1 at the end of the response letter) to the revised manuscript.

(3) The limited growth of vegetation on higher elevated region can be because of several reason: persistent snow cover/ no loose materials to grow the roots or bare rocks / active landslides – high erosion / climate (rainfall and temp.). It is important to investigate these areas in detail using google earth images and deepen your discussion section.

Response: Thanks for your suggestions. We explored these places in the Google Earth and added some discussions to explain the slow vegetation recovery on top of the high mountains.

"Our finding that elevation is the most important influencing factors indicate that the slow vegetation recovery on high mountains may be controlled by cold harsh weather or persistent snow cover in winter seasons. We also explored very high spatial resolution images on the Google Earth platform and found absent of vegetation on both landslide scars and deposits. In warm humid climate, primordial plants such as lichen and moss may grow and lead to increased EVI. Therefore, we hypothesize that either the climate on high mountains or remobilization of these landslides inhibited vegetation recovery on landslide surfaces."

(4) Figure 2. it is quite misleading by showing high values of annual EVI increasing rate and then understands it is to multiply by 10Ё̈Ę-3. Is this increasing rate statistically significant ?

Response: As we used a linear model, the model derived EVI on the 15th July of post-seismic years is changing linearly and statistically significant.

(5) Figure 3 C. How do authors calculate the landslide depth ? (source).

Response: The landslide depth is a rescaling of the landslide area by using the equation in Xu et al. (2016, Scientific Reports, 6, 29797). We added a few descriptions to make it clearer.

(6) Figure 3 L : basically all the type of trees/plants is damaged after the earthquake . So the comparison shown in Fig 3 L is meaningless

Response: We deleted this part of the analysis.

(7) Author wrote "We found >99% landslide surfaces have been recovering since 2008" . Where did this value comes from ?.

Response: We made a histogram of annual EVI increasing rate as an inset of Figure 2a. The histogram clearly shows that ">99% landslide surfaces have been recovering".

(8) Section 4.4 is not a proper discussion.

Response: This part has been deleted.

(9) I am not qualified to judge the English, but I feel there is considerable scope to improve. For eg., Line 25 – 'thousands of hundreds'

Response: Grammar of the manuscript has been checked and language editing services may also be used in later version of the manuscript.

$$EVI_x = a_0 + a_1 \cos\left(\frac{2\pi}{T} x\right) + b_1 \sin\left(\frac{2\pi}{T} x\right) \qquad \text{(Eq. 1)}$$

where,

$x$ Julian date, $a_0$ constant to be estimated, $a_1$ $b_1$ coefficients for intra-annual change for EVI.

**Fig. 1.** Eq. 1